# Mental Sleep Activity and Disturbing Dreams in the Lifespan

**DOI:** 10.3390/ijerph16193658

**Published:** 2019-09-29

**Authors:** Serena Scarpelli, Chiara Bartolacci, Aurora D’Atri, Maurizio Gorgoni, Luigi De Gennaro

**Affiliations:** 1Department of Psychology, “Sapienza” University of Rome, Via dei Marsi, 78, 00185 Rome, Italy; serena.scarpelli@uniroma1.it (S.S.); chiara.bartolacci@uniroma1.it (C.B.); aurora.datri@gmail.com (A.D.); maurizio.gorgoni@uniroma1.it (M.G.); 2IRCCS Santa Lucia Foundation, 00142 Rome, Italy

**Keywords:** dreaming, lifespan, sleep, cognition, children, young adults, elderly, nightmares, PTSD

## Abstract

Sleep significantly changes across the lifespan, and several studies underline its crucial role in cognitive functioning. Similarly, mental activity during sleep tends to covary with age. This review aims to analyze the characteristics of dreaming and disturbing dreams at different age brackets. On the one hand, dreams may be considered an expression of brain maturation and cognitive development, showing relations with memory and visuo-spatial abilities. Some investigations reveal that specific electrophysiological patterns, such as frontal theta oscillations, underlie dreams during sleep, as well as episodic memories in the waking state, both in young and older adults. On the other hand, considering the role of dreaming in emotional processing and regulation, the available literature suggests that mental sleep activity could have a beneficial role when stressful events occur at different age ranges. We highlight that nightmares and bad dreams might represent an attempt to cope the adverse events, and the degrees of cognitive-brain maturation could impact on these mechanisms across the lifespan. Future investigations are necessary to clarify these relations. Clinical protocols could be designed to improve cognitive functioning and emotional regulation by modifying the dream contents or the ability to recall/non-recall them.

## 1. Introduction

In the last decade, studies on mental sleep activity have shown remarkable advances. Dream experience has been considered an expression of consciousness during sleep [1], although the detection of the specific instant in which mental activity is produced could be challenging. As is well known, dream research is characterized by a notable limitation: the existence of a temporal gap between dream generation and the moment in which the dream is reported. In the attempt to overcome this “asynchrony”, some authors have used the eye-signaling technique, i.e., pre-decided sequences of intentional eye movements during dreaming [2,3,4], to reveal the exact moment of dream production. However, while this method is successfully applied to lucid dreams, it is not applicable to the “traditional” mental sleep activity. The access to dream contents is mainly obtained from the information reported by subjects after awakening.

Furthermore, multiple variables affect the possibility of obtaining a dream recall, such as the method of collection, the sleep stage upon which subjects are awakened, and the macrostructural and microstructural features of the sleep interval before dream recall (for a review, see [5]).

One of the best-established theories on dreaming is the “continuity hypothesis”, firstly proposed by Hall and Norby [6]. The authors hypothesized that thoughts, behaviors, fantasies, and emotions experienced during the waking state may have their continuity in sleep. Several investigations provided empirical support for this hypothesis [7,8,9,10]. In particular, some results underlined that specific neurobiological circuits are responsible for both cognitive and emotional experiences during the waking state as well as mental sleep activity. Hence, the dysregulation of these mechanisms can affect both daytime and dream experience (for a review, see [11]). In this view, the observations by Solms [12,13] suggested that specific brain injuries affect some cognitive functions during dream imagery as much as wakefulness. For instance, “visual anoneria” was related to lesions in the visual associative cortex (ventromesial occipito-parietal cortex) and produced dream imagery with a total or partial drop in the production of visual contents. In parallel, this condition is associated with compromised visuo-spatial skills and the ability to produce mental imagery during wakefulness [13]. Another condition described by Solms [13] is “anoneirognosis”, characterized by the inability to discern dream from reality, due to lesions over medial prefrontal cortex, anterior cingulate cortex, or basal forebrain. In these patients, a general state of confusion during the waking state and dream contents can intrude into their waking thoughts [9]. 

Behind the relation with cognitive functioning, it should be underlined that several regions involved in emotional regulation during wakefulness are also activated during rapid eye movement, (REM) episodes associated with dream recall [14]. In particular, the limbic system and the reward dopaminergic system activation associated with dream experience during REM sleep can promote emotional processing and learning [15,16]. Recent neuroimaging findings point in this direction, showing that microstructural measures of specific limbic areas were related to qualitative and quantitative features of dream reports [15,16,17,18]. Moreover, subjects with high dream recall frequency showed greater regional cerebral blood flow (rCBF) than low recallers in the temporo-parietal junction (TPJ) during REM sleep, stage 3 NREM sleep, and wakefulness, and in the medial prefrontal cortex (mPFC) during REM sleep [19]. Both TPJ and mPFC have a crucial role in emotional and cognitive processing during wakefulness (for a review, see [20]). Specifically, the TPJ is involved in the theory of mind, empathy, and social cognition during the waking state, while the mPFC is related to awareness, introspection, attention, and identification of emotions [20].

This neurobiological evidence substantially supports the notion that some neural substrates are shared between waking emotional/cognitive functioning and dreaming. However, findings addressing how these mechanisms change across the lifespan are still lacking. Another issue concerns the role of dreaming in specific disorder, such as post-traumatic stress disorder (PTSD) and idiopathic nightmares, where dreams are abundant and contents are frequently emotional and/or bizarre (for a review, see [11]). It is still unclear whether dreams only reflect the alterations related to these disturbances or play a particular role in these pathological conditions at different ages of life.

The current review aims to provide a comprehensive overview of these issues, including, for each age bracket, a summary of the micro- and macrostructural characteristics of sleep, the dream features, and the available findings on disturbing dreams. We have focused our review on nightmares and PTSD, in which the alterations of the mental sleep activity are relevant. We considered the following age ranges: (a) children and early adolescence (29 months to 16 years); (b) young adults (18–47 years, but individuals up to 60 years were part of the clinical sample in some studies (e.g., [21]); (c) older adults (50–94 years). 

## 2. Dreaming Across the Lifespan

### 2.1. Children

#### 2.1.1. Sleep Pattern Development and Cortical Maturation

Sleeping could be seen as the main activity of a neonate in the first period of life. Since the first hours after the birth, sleep in newborns shows two different states with distinct behavioral and cortical activity patterns: active sleep (AS), representing the antecedent of REM sleep, characterized by rapid eyes movements, a quite stable electroencephalographic (EEG) background of mixed rapid, low amplitude theta and alpha activity, and mainly synchronous occipital delta activity, and irregular heart rate and respiration, associated to facial and head movements [22,23,24]; and quiet sleep (QS), antecedent of the non-REM (NREM) sleep, with high amplitude, mixed frequency EEG, and tracé alternant EEG activity, associated with a regular cardiorespiratory pattern and few body movements [22,23,24].

During the first weeks of life, signs of AS and QS could occur simultaneously, during the so-called indeterminate-sleep (IS) that progressively disappears [25], pointing to the stabilization of the sleep controlling processes. In newborns, the sleep period starts with active sleep, while the entrance into the sleep cycle through NREM sleep begins at 3–6 months. 

The first months of life represent a critical period for brain development. The maturational trajectory of cortical areas sees the primary sensory-motor cortices developing first, and then it gradually shifts anteriorly and laterally, involving areas linked to more complex associative functions [26]. The maturation of the central nervous system is paralleled by the QS and AS stabilization and the gradual development of features characterizing the mature NREM and REM sleep, respectively. 

The first sign of this sleep maturation is represented by the appearance, at two months of age, of the first sleep spindles during the QS/NREM sleep, whose occurrence reaches a peak during the second half of the first years of age and then declines [27,28]. Sleep spindles are phasic events within the sigma frequency band (11–15 Hz) generated within the thalamo-cortical system [29]. A boost in spindle density [27,28] could be related to functional changes during this period occurring within the thalamo-cortical circuit. Spindle activity is involved in long-term memory consolidation [30,31] and cognitive abilities [32]. Accordingly, the developmental evolution of spindles might represent an index of maturation of the thalamo-cortical network as well as of the cognitive functions related to its functioning. 

At about six months, the k-complexes, phasic events with high amplitude and low frequency (<1 Hz), likely generated at cortical level [33,34,35,36,37], also appear in the NREM sleep EEG [1,18]. Furthermore, quantitative analyses of the sleep EEG during infancy, childhood, and adolescence show that cortical activity during QS/NREM sleep undergoes substantial changes mirroring the cortical maturation processes [28,38,39,40,41,42,43,44,45,46,47]. Major changes in quiet/NREM sleep activity during infancy and childhood to adolescence mainly involve the low-frequency bands, with earlier changes reflected by the theta activity [27,38,45], and the subsequent modifications mainly involve slow wave activity (SWA) [42,45,48]. These modifications might be summarized as a posterior-to-anterior shift of the theta/SWA prevalence, which parallels the trajectory of the concurring structural and functional cortical maturation [38,42,49]. The active/REM sleep at birth represents the sleep onset stage and 50% of sleep in a newborn. This percentage progressively decreases to reach the level of 20–30% at around five years of age [50]. This pattern of changes points to a still undefined special role played by REM sleep in infancy. It has been proposed that AS/REM sleep in the developmental period provides constant activation inputs guiding the cortical plasticity process, overexpressed during the synaptic proliferation period, when the system does not receive external sensory stimulations from the environment [51]. Changes in cortical activity during REM sleep also involves delta/theta oscillations [40,45,52]. An initial increase in EEG power, reaching its peak at two years, is then followed by a gradual decrease persisting to adolescence [27,52,53]. Bursts of oscillatory events within the fast delta/slow theta frequency have been reported to occur during REM sleep in the second year of life, and to subsequently decrease with increasing age, reaching the adult level by around five years [45]. 

Given the frequency- and regional-specificity of these changes in cortical oscillations during sleep and the changes in functional connectivity accompanying them [48,49,54], some authors also proposed that these modifications in cortical rhythms play a key role in cognitive development and that sleep, by mediating and driving the cortical plasticity processes in critical periods, promotes the acquisition of new abilities, from motor-sensory to more complex cognitive and socio-emotional functions, such as self-awareness, semantic memory, response inhibition, and self-regulation strategies [55,56,57,58,59,60,61,62,63]. 

#### 2.1.2. Dream Experience During Childhood

The ability to recall dream experience develops from 3–4 years, since children cannot organize and retrieve mnestic traces below the age of three, resulting in the so-called “infantile amnesia” [64,65]. Furthermore, some studies have revealed that children begin to distinguish dream from reality at four years, but they can completely discern the two phenomena only at nine years [66,67,68,69,70].

It has also been observed that some dream contents usually present in healthy adults, such as people or objects, are missing in dream reports of children from three to five years of age [66]. Children start to show interest in their mental sleep activity at six years [71] and to represent themselves in dreams at the age of seven years. At the same time, dreaming becomes characterized by high involvement and emotional charge [66,68,69]. Dream reports similar in complexity to those of adults are collected around nine years of age, and the first longer reports are reached from five to seven years of age [66,67,68,69,70]. Not surprisingly, in the same period, the shift in the SWA peak during sleep reaches the frontal areas, as an index of structural and functional cortical maturation [42,49], and the cortical oscillations during REM sleep stabilize [45].

However, very few studies during childhood provide information on dreaming in healthy subjects, and most of the data were obtained from clinical samples (see next Section) [72,73]. A notable exception was represented by a study focusing on healthy children [66,74], showing that children aged seven years produced a lower percentage of dream recall than adults. In particular, children awakened from sleep in a laboratory setting have a 20% dream recall rate, while in adults, it is 80–90% [74]. Consistently, investigations by retrospective methods observed that dream recall frequency is lower in children [75].

The fact that dream production seems to be scarce or absent in children younger than three years supports the view that dream features could be an expression of cognitive development [76]. In this vein, performance in some neurocognitive tasks has been studied in relation to dreaming. The available results on this issue revealed no correlation with language measures, while visuo-spatial abilities and organizational skills were strongly linked to dream recall [66,74,76]. In particular, the Block Design Test [77] showed the highest correlation with dreaming. Not surprisingly and partly in line with Solms’ observations [13], these results revealed that executive functioning is related to dream recall.

The hypothesis that mental sleep activity parallels cognitive maturation was also directly tested [78]. The author assessed the correlation between the bizarreness of oneiric contents and cognitive abilities in children aged 3–5 years. The study was carried out with home and kindergarten interviews that replicated the findings on the relation between visuo-spatial abilities and dreaming. Specifically, dream bizarreness seems to be correlated to cognitive functions, such as linguistic skills, long term memory capacity, attention span, symbolization, and visuo-spatial skills [78].

More recently, Sándor et al. [79,80] tried to disentangle the issue of the relationship between executive function and dream features [80]. Inhibitory control, emotional interference control and some dimensions of executive functioning in a situation of interfering stimuli were tested in 40 healthy children (4–8.5 years). Furthermore, different measures of attention were collected, along with the assessment of verbal ability, visuo-spatial ability, and working memory [80]. The authors found that the number of human characters per dream report is related to the ability to select the relevant stimuli in a distracting environment. Moreover, the rate of self-initiated actions and gross-motor activities in dream reports was significantly related to a better inhibitory control [80]. For what concerns the linguistic competence, the performance was positively correlated to the number of verbal actions per dream [80]. 

Overall, these findings confirmed, to some extent, that dream characteristics are actually related to executive functioning and waking skills. However, it should be noted that the relation between dreaming and visuo-spatial abilities [66,74,78] was not replicated by Sándor et al. [80].

#### 2.1.3. Disturbing Dreams in Children

As previously mentioned, most of the studies investigating the children’s oneiric activity focused on the presence of nightmares, which are defined as disturbing dreams with negative emotions, such as fear and vivid and realistic images [81,82,83,84].

Several studies showed that the presence of bad dreams and nightmares is quite common from pre-scholar age to 14 years [82,83,84,85]. These studies showed high percentages of children reporting bad dreams with a frequency of at least once per month or six months, reflecting a relatively stable characteristic that seems to emerge during the first years of life [86]. Furthermore, idiopathic nightmares, i.e., without a known cause, begin during early childhood and decrease in adulthood, and they are more prevalent in females [87]. It was shown that idiopathic nightmares result from dysfunction in the hippocampal–amygdala prefrontal circuit that controls fear memory formation and extinction [88,89]. Nevertheless, some studies demonstrated that idiopathic nightmares have comorbidity with anxiety, depression, and insomnia [90], and they should be differentiated from recurring nightmares in PTSD [91,92], which develop in consequence of a history of trauma [93]. However, a recent investigation in which mothers were prospectively requested (at 2.3, 3.5, 4.8, and 6.8 years) to collect their children’s oneiric activity found that the rate of nightmares was higher in relation to abuse history and family adversity occurring at 0–4 years [94]. This finding supports the hypothesis that nightmares, as a consequence of such adversity, could share pathophysiological mechanisms with PTSD nightmares [87]. Adults suffering from idiopathic nightmares were also exposed to traumatic-like events during the infancy and early childhood and showed specific alterations in EEG sleep pattern, i.e., lower slow sleep spindle.

A study examining the relation between nightmares and other sleep problems during childhood (2–15 years) showed that disturbing dreams are more frequent in children aged 6–10 years. In addition, nightmares are higher in children with circadian rhythm disorders during the first year of life (46%) as compared to regular sleepers (29%) [85]. On one side, sleeplessness symptoms could predict the presence of bad dreams in young children. Conversely, a longitudinal study on parent-rated disturbing dreams at 29 months, 41 months, 50 months, 5 years, and 6 years [86] revealed that sleeplessness did not predict bad dreams at any age, suggesting that such relationship between nightmares and insomnia in adults would appear later during the development. However, the association between nightmares and psychiatric symptoms seems to be confirmed in childhood. In fact, children with emotional disorders and generalized distress, such as separation anxiety [95] or difficult temperament in infancy [96,97], more frequently report bad dreams [95,98]. Furthermore, comorbidity between insomnia disorder and nightmares in adolescents has been reported [99,100]. Interestingly, Simard et al. [86] showed that the best predictor of bad dreams was the presence of disturbing dreams at the preceding age, suggesting that this may be a stable trait of subjects. Bad dreams at 29 months predicts disturbing dreams at 5–6 years, i.e., a period of life characterized by several challenges that may result in high level of stress. In this view, the child’s vulnerability to stress at 29 months can interact with stressful life changes (e.g., starting school) [86]. 

It is important to specify that nightmares must be distinguished from sleep terrors (night terrors or pavor nocturnus), defined as a NREM parasomnia belonging to the disorders of arousal [101]. Generally, sleep terrors appear during childhood (34% of children at 1.5 years of age), in association with confusional arousals and sleepwalking, and resolve by puberty [102]. Although the prevalence of sleep terrors during childhood has never been accurately assessed, they are highly prevalent in children between the ages of 3 and 13 years [103]. In particular, sleep terrors are characterized by an arousal activation in stages 3 or 4 of NREM sleep and intense fear with different physiological alterations, like tachycardia, sweating, and mydriasis. The child shows himself shaken, not responsive, inconsolable, and, after the awakening, he cannot remember the episode [81]. These features differ from nightmares, in which the fear is less intense and the child is able to remember the dream content. LaBerge et al. [103] found high anxiety scores in children that suffer from night terrors, while Rosen et al. [104] hypothesized that children with night terrors would show a reaction to significant and aversive life events, such as parents’ separation or family conflicts. 

Although sleep disorders in children are pretty common, they are still poorly investigated, and studies about the related dream experience are very scarce. They are more likely to appear in concomitance of inadequate sleep hygiene [81], for which insufficient quality of sleep represents an important factor, contributing to the development or maintaining of nightmares, bad dreams, or parasomnic episodes. In addition, they occur with higher probability as a consequence of stressful events, like exclusion from a group [105], parents’ divorce [106], or loved people physically ill or passing away [105]. These findings partly support studies in adults that demonstrate a contribution of dreams to emotional regulation [89,107].

### 2.2. Young Adults

#### 2.2.1. Sleep Stabilization and Cognitive Functioning in Adults

In late adolescence, different brain regions complete the maturational processes, with the higher-order association areas concluding their development last [26]. The areas needing the longest maturational process are parts of the temporal cortex and the dorsolateral pre-frontal cortex, which keeps developing until the end of the adolescence [26]. Actually, the maturational sequence underlies the gradual acquisition of increasingly complex cognitive functions, starting from the simplest sensory-motor, passing through the development of spatial orientation, language, and speech abilities, and ending during adolescence with the development of the executive functions [26]. 

The cortical maturation leads the stabilization of the sleep features, showing all the typical EEG signs of adults. Sleep periods mainly occur during night-time, with a delay in the bedtime as a function of age [108]. Sleep becomes more continuous with a reduction in the number and duration of intra-sleep awakenings [109]. The sleep onset occurs by entering in NREM sleep, and the NREM-REM switch, which defines the sleep cycles, encompasses the whole night period. The transition from childhood to adolescence is associated with a modest but significant increase in the amount of REM sleep and a decrease in the percentage of slow wave sleep (SWS) [109]. During adolescence, the reduction in the grey matter, following the initial growth during the early infancy, is associated, during sleep, to a marked reduction of cortical oscillation power and amplitude [39,41,110,111]. The posterior-to-anterior shift in cortical maturation, which is mirrored by the consistent SWA-maxima shift during NREM, completes its course [42,47,112]. Unlike grey matter volume, white matter volume grows during adolescence. The improvement in cortical connectivity is reflected during sleep by the increased EEG coherence, encompassing all sleep stages and frequency bands [113], and pointing to a general improvement in cortical communication that underlies the new and more complex cognitive functions.

The strong relationship between EEG activity during sleep, structural and functional brain changes, and cognitive functioning suggests the possibility of investigating adult cognitive functioning by looking at specific trait-like features of sleep electrophysiology. For instance, sleep spindle activity shows a positive relation with the intelligence quotient (IQ), which is stronger in females than in males [114]; reasoning and learning abilities; and visual-spatial working memory [115,116,117,118,119]. On the other hand, SWA during NREM sleep seems to locally reflect the experience-dependent plasticity that occurred during wake (e.g., [120]) and plays an important role in sleep-dependent memory consolidation. Accordingly, several studies showed that the sleep-dependent overnight gain in perceptual and visuo-motor learning, working memory tasks, declarative memory, and, in general, hippocampal-dependent learning tasks is mediated by SWA [121,122,123,124,125,126].

Briefly, it has been proposed that the repeated switch between NREM and REM sleep in consecutive cycles consolidates learning. During the NREM-REM sleep sequences, two complementary processes would occur: (1) the downscaling of cortical plasticity in a learning-dependent way though SWA [127], and (2) the connections strengthening within specific circuits through their reactivation (replay) during higher frequency activities, such as spindles [128,129] and hippocampal ripples [129,130,131] or during REM sleep [128,132]. These processes lead to the increase in the signal-to-noise ratio of cortical communication resulting in learning [133]. Moreover, REM sleep seems to be also implicated in the processes of assimilation and generalization of the new memories into a common schema underlying the generalized knowledge and creativity [133]. This metalevel role of REM sleep has also been linked to the features of mental activity during sleep, potentially explaining the bizarreness and the lack of concreteness usually associated to dreams [134]. 

#### 2.2.2. Neural Bases of Dream Recall in Young Adults

In keeping with research on healthy children, some studies investigated the relationship between cognitive measures and dream features in young adults. Interestingly, bizarreness seems to be the only qualitative parameter that appears to be correlated with cognitive processes and skills in young adults studies as well as in children [78,135]. Visuo-spatial skills were assessed in relation to dreaming, showing that the dream recall frequency upon awakenings from REM sleep was associated with higher performance in the Block Design Test [135]. Moreover, in another investigation, the gain in the Mirror Tracing Task—assessing visuo-spatial motricity and eye-hand coordination—was better in high than low dream recallers [136].

More recently, studies on dream recall in young adults used the polysomnographic (PSG) recordings with provoked awakenings (e.g., [9,137,138,139]). Specifically, the electrophysiological background of the last segment of sleep before the awakenings seems to predict the subsequent presence/absence of dream recall. Interestingly, theta activity, besides having a crucial role in the cortical maturation [38], represents one of the main EEG signatures that predicts dream recall [8,9]. Both within- and between-subject investigations revealed that the last five minutes of REM sleep with high frontal theta oscillations are related to the retrieval of dream content upon awakenings [8,9]. Moreover, the occurrence of the EEG theta activity associated with dreaming does not depend on trait-like characteristics of subjects recorded but does depend on state-like factors [9,140]. 

Further results revealed that theta activity in university students during REM sleep was also related to the incorporation of daytime experience in dreaming [10]. Frontal theta activity correlated to the rate of waking experiences referred by the subjects into dream reports. These dream contents included a significant percentage of emotional items [10]. The results are substantially consistent with the view that frontal theta activity during REM sleep is associated with the consolidation of emotional memories [141,142]. At the same time, robust evidence points to the relation between the frontal theta activity and cognitive processing also during wakefulness. Memory encoding and retrieval for episodic information are linked to theta oscillations (for a review, see [143]). Theta oscillations are strictly related to hippocampal activity, promoting memory formation and encoding, as confirmed by intracranial-EEG (iEEG) studies [144,145]. Besides, theta activity modulates the interaction between other regions involved in the recall processes such as medial temporal lobe and mPFC [146].

The alpha EEG band also seems to be involved in dream recall, albeit the results are more heterogeneous concerning the topographic distribution. Two studies found that alpha activity is lower in parieto-occipital regions during stage 2 NREM sleep in association with successful dream recall [8,147]. Furthermore, Esposito et al. [147] found a reduction in alpha power also over the right frontal regions during NREM, and these effects were replicated in REM sleep. Differently, a protocol with multiple naps in 40 hours revealed that high occipital alpha activity is related to dreaming during REM sleep [148]. Consistently, another investigation on awakenings from sleep onset REM periods showed that increased alpha over the central area was associated with the absence of dream recall, while it was related to dream experience from NREM periods [149]. 

Interestingly, fluctuations in alpha power are also related to memory performance during wakefulness [150]. Moreover, the alpha activity has been proposed to mirror the inhibition of cortical activity not involved in the ongoing brain operations during waking state [150]. In addition, the so-called phenomenon of "Frontal Alpha Asymmetry" (FAA), i.e., the difference in frontal alpha activity between the two hemispheres, is considered as an index of affective regulation during wakefulness [151]. Sikka et al. [152] found that FAA was positively correlated with affective experiences in REM sleep dreams, especially with dreams containing “anger”. In other words, subjects with lower right frontal activity expressed in terms of high alpha power seem to be less able to inhibit strong affective states during sleep/dreams. 

It should be noted that, while the theta activity has been suggested as being state-dependent, the alpha power is more likely associated to trait-like factors [153].

Recent findings have revealed that high-frequency EEG activity is also related to dream experience. In particular, an increased gamma power predicts dream recall [138,154]. Moreover, dream recall during NREM sleep has been related to lower slow waves [137,138,155] and less fast spindles in correspondence of central and posterior cortical areas [155]. These findings highlighted that dream experience is possible when the brain revealed a certain degree of activation and are in line with the hypothesis that the higher cortical arousal and light sleep promote the encoding and consolidation of the dream experience during sleep [137,156,157].

Although these results could appear conflicting with those on the theta/alpha activity, it should be noted that waking EEG studies point to an interaction between theta and gamma oscillations (for a review, see [158]). Specifically, the so-called theta/gamma coupling seems to be particularly involved in long-term memory encoding [158], also promoting the retrieval of stored mnestic traces in animals as well as in humans [159]. 

#### 2.2.3. Disturbing Dreams in Young Adults

Nightmares during adulthood are associated with impoverishment in subjective sleep quality and daytime consequences on well-being (e.g., [160]), such as psychological distress, depression, and anxiety symptomatology [90].

A recent study highlighted that the activation of the autonomic nervous system could be associated with nightmares [161]. Paul et al. [161] hypothesized that nightmares, as a kind of psychological stressor, would activate two different autonomic responses. They found an increase in heart rate frequency, breathing cycle length, electrodermal responses, and REM sleep density, showing increased autonomic activity in people who experienced nightmares [161]. These results are consistent with the assumption that high emotional charge in dreams results in increased arousal [162]. Interestingly, Simor and colleagues [163] analyzed the cycling alternating pattern (CAP; [164]), an index of sleep stability, and found that the NREM sleep of university students with frequent nightmares is characterized by a reduced A1 phase of CAP, i.e., a parameter linked to EEG synchronization in sleep, and by enhanced A2 and A3 phases of CAP, which reflect increased high-frequency EEG activity. Overall, a greater autonomic and electrophysiological activation can contribute to the occurrence of nightmares.

Besides, various studies on adult subjects demonstrated that some electrophysiological features that are REM-sleep-related would play a pivotal role in emotional processes [89]. Recently, it was suggested that the EEG frontal theta activity during REM sleep might modulate dysphoric dreams, as nightmares [89], altering the function of the amygdala and increasing abnormal oneiric activity in PTSD patients [165]. The presence of higher slow theta activity (2–5 Hz) in the frontal and central areas has been observed in a sample of nightmare recallers compared to controls. Moreover, Spoormaker and colleagues [166] reported that the suppression of REM sleep would impair fear extinction and that the percentage of REM sleep obtained in consequence of fear extinction can predict decreased arousal as measured by skin conductance [167]. Finally, although no dream measures were collected, some authors suggested an association between higher frontal theta activity and emotional coping strategies in resilient subjects that did not ever suffer from PTSD after a trauma exposition [168]. However, the lack of dreaming investigation in this study does not allow clarification as to whether the adaptive role of the theta activity is related only to REM sleep or, generally, to the mental sleep activity. It is also interesting to consider that episodes with high emotional load or stressful events can be incorporated into dream experiences during the days after the events until several years later [169]. In this regard, we could speculate that dreaming would re-activate waking-life experiences to re-process negative emotions of conflictual events [107]. Vallat and colleagues [170] reported that the presence of waking-life experiences in dream recall coincides with a lower emotional load, supporting the hypothesis that dreaming promotes emotional regulation and reduces the emotional load of wakefulness memories. 

Nightmares can also frequently occur in narcoleptic patients [171,172,173], with a percentage of 33% [173] as compared to the prevalence of 5% reported in the general population [174,175]. These patients usually experience very vivid, bizarre, and frightening dreams [176], with frequent aggressive scenes [177]. In this respect, it is important to specify that one of the main symptoms in narcolepsy, such as the cataplexy, is usually triggered by events with a high emotional charge [178] and that such patients show an impairment in the limbic system (e.g., [179]). For these reasons, to some extent, narcolepsy, emotional processes, and dreaming are inter-connected. Disturbed emotional contents in REM sleep dreams of narcoleptic subjects may be higher because of the impairment of cognitive-emotional processing and regulation [180]. 

It is well known that narcoleptic patients reported many lucid dreams compared with controls [181]. It was proposed that sleep features of narcoleptic patients, such as short sleep latency, anticipated REM onset, and fragmented sleep, and facilitate the appearance of lucid dreams [182]. Consistently, some studies induced lucid dreams in healthy individuals (aged 18–60 years) through methods promoting cortical arousal [154,183]. In this regard, albeit lucid dreams were not collected, a recent study found a relation between higher brain activation, i.e., lower values in the delta/beta ratio, and dream recall in narcoleptic people [21]. 

Interestingly, it was suggested that dream lucidity may help nightmare sufferers intervene on their dream contents, modifying the negative emotional load of the dream experience [4,184]. Rak et al. [182] revealed that narcoleptic subjects had relief when they experienced nightmares with lucid dream episodes. In this respect, lucid dreaming could be considered as a sort of coping strategy with unpleasant emotions in dream experience [185], and some authors proposed the induction of lucid dreams as a useful intervention for nightmare disorders [186].

### 2.3. Aging

#### 2.3.1. Sleep Pattern in Aging

The stability and regularization in sleep features and dynamics reached at adulthood tend to deteriorate during the fifth decade of life. Changes in sleep associated with aging involve both macro- and microstructural features, often in the opposite direction relative to those observed during development. The time at which older adults go to bed and wake up advances, the total sleep duration reduces, and sleep becomes more fragile and fragmented, with more frequent intra-sleep awakenings or arousals [109,187]. The NREM-REM sleep cycle progressively shortens and the number of cycles during the night reduces [187]. On the other hand, the time spent sleeping during the daytime increases, with about half of the diurnal naps being unplanned in older people [188]. The most stable and evident variation associated with aging is the reduction of the deepest SWS and its substitution by lighter NREM stages 1–2 [109,187]. Findings on quantitative EEG changes during NREM sleep showed consistent results by reporting significant reductions of SWA from young to middle-age [189,190,191] and a further prominent SWA impairment in older people, reaching a 75–80% of average decrement compared to younger adults [192]. These changes are most prominent over the prefrontal regions [187,189,190,191,192] and also affect features other than EEG activity. Indeed, amplitude and density of SWA progressively decrease from young to middle-age [193] and keep reducing in older age [194], with the main variations affecting the first 1–2 cycles [193]. Furthermore, the steepness (slope) of the slow wave becomes shallower [193], pointing to an impairment in the ability to synchronize neuronal firing in large populations of neurons, i.e., in the mechanism generating this kind of oscillations [195].

Sleep spindles, whose functions and features are strictly linked to those of SWA, follow a similar pattern of changes during normal aging. A significant reduction of spectral power within the 12–15 Hz spindle range has been reported in middle-aged and older adults [29,189,190]. In the same vein, the number (density), amplitude, and duration of sleep spindles also decrease in normal aging, with density and amplitude maximally changing on the frontal areas [29,196,197], while major variations in spindles durations are reported on the parieto-occipital regions [197]. Unlike modifications in SWA, the disruption of these measures of sleep spindles mainly affects the last cycles of the sleep period [29,196,197].

On the one hand, this inversion in the direction of sleep changes passing from childhood-adolescence to adulthood-middle-age-elderly has been related to the inversion of structural and functional changes occurring in the brain. The gradual deterioration of the sleep-wake regulatory system [198,199] and the age-related thinning of the gray matter [192,194,200] have been associated with sleep instability and fragmentation, and with the impairment of brain oscillations reported in the elderly. Accordingly, these changes can be considered symmetrical to those seen during the developmental period, when the development of sleep controlling circuits and the maturation of cortical regions during the first years of life improve sleep continuity and the ability of the brain to express cortical oscillations during sleep. On the other hand, according to the critical role of these sleep oscillations in memory processes, it is not surprising that their impairment has been linked to the cognitive and memory decline observed in normal aging [187,196]. 

Only few studies reported REM sleep modifications in older adults. Its percentage in nighttime sleep decreases until 60 years of age [109]. Only one study reported microstructural changes in REM sleep in normal aging, while major frequency- and regional-specific modifications in this sleep stage have been associated with several neurodegenerative disorders, such as Alzheimer’s disease [201,202]. EEG activity in low-frequency bands (delta, theta, alpha) has been reported to decrease in one study [203]. Looking at the regional-specificity, the cortical activity in the alpha and beta1 frequency range seems to decrease mainly at the centro-occipital areas, while the theta activity power has been reported to increase in frontal regions with age [203]. However, given the very low sample size in this study (seven young vs. seven old adults), these findings should be confirmed. 

#### 2.3.2. Dream Recall and Disturbing Dreams during Aging

The main results about dream experience in older people revealed a critical drop in dream recall frequency [204,205], likely linked to the reduction of REM sleep. Several authors hypothesized that the changes in dream recall rate [64,204] could also be considered an expression of age-dependent cognitive deterioration. Some studies on older patients provided support to this hypothesis [206]. Specifically, people suffering from mild cognitive impairment showed both reduced REM sleep and dream recall rate compared to healthy subjects [206].

An earlier study by Kramer and Roth [207] explored dream recall in relation to age and severity of damage in hospitalized male patients with a clinical diagnosis of chronic brain syndrome (mean age of elderly group = 70 years). It was found that severely damaged patients had a lower dream recall rate than mildly damaged, and, in particular, most of the severely damaged older people had a complete loss of dream experience. The authors stated that both aging and severity of brain syndrome could affect dream recall. They also considered two hypotheses: (a) the drop in dream recall may be potentially due to the absence of experiences that patients can translate into verbal terms; (b) the loss of dreaming can be explained by the failure in memory processes [207].

However, other studies showed that this age-related reduction in dream recall seems already reduced in first and middle adulthood [64,204,208], and no differences occurred between middle-aged and older adults [64,209]. Hence, other explanations are necessary to clarify the relationship between aging and dreaming.

A study on dreaming in a large sample aged 10–79 years revealed that some dream features change across the lifespan [64]. The results showed a linear reduction in the number of dream themes across ages. The decrease in the oldest group (60–79 years) was significant not only when compared with the youngest group (10–19 years) but also with respect to the group aged 50–59. According to the view that the dream theme diversity could be considered as an index of episodic and autobiographical memory functioning [64], its reduction across the lifespan may be associated with the decline in autobiographical memory during aging [64]. 

Furthermore, dream features other than theme diversity, such as dream length, seem to be related to some cognitive domain [209]. Specifically, a positive correlation between visual memory performance and dream length has been found in a study assessing quantitative dream features in people from 61 to 75 years. 

Along this vein, more recent studies in older adults assessed the relation between dreaming and autobiographical memories [210,211]. In order to understand whether temporal references identified in dream reports follow the same temporal distributions as those revealed for autobiographical memories, Grenier et al. [210] recorded, in a laboratory setting, both younger women (18–35 years of age) and older women (60–77 years of age), and collected dreams at the morning awakenings. Subjects were required to identify temporal references in their reports and produce a sample of autobiographical memories using the semantic cuing method. Both groups reported a linear reduction in the temporal references identified in dream recall and autobiographical memories with increased remoteness for the last 30 years [210]. In particular, the older group showed that the distribution of dream temporal references followed the typical distribution of waking autobiographical memories, characterized by a significant amount of references concerning the last 10–20 years and the adolescence or young adulthood experiences, i.e., the reminiscence bump [210,211,212]. It should be underlined that these findings provide strong support to the continuity hypothesis between waking and dreaming memory processes. Furthermore, the analysis of the dream contents of the 30 older women involved in this protocol [210] provided a possible explanation on these “reminiscence bumps” [211]. The presence of a greater percentage of content referred to adolescence and young adulthood may be related to the salience of experiences contributing to the development of self-identity and life goals [211]. Consistently, some studies revealed that the reduction in dream salience in older people is related to lower dream recall rate [213].

Interestingly, another study assessing dream reports of 100 elderly people (mean age = 77 years) from nursing homes showed that their contents included a greater amount of familiar and friendly characters and positive emotions from a “happy past” [214]. The authors suggested that these autobiographical memories reported in mental sleep activity help older people deal with the present stressful/uncomfortable situation in a nursing home [214].

Only few investigations have been performed on neural correlates of dreaming in older adults. Chellappa et al. [215] found that higher delta and sigma power, respectively over frontal and centro-parietal areas, predicted dream recall in elderly people (57–74 years) during NREM sleep. No differences were found for what concerns awakenings from REM sleep. These findings are scarcely consistent with the available literature on dreaming. However, the multiple awakenings obtained in a protocol of 40 hours under constant routine might have affected the results by providing a sort of “sleep deprivation” in older subjects, therefore causing the increased delta activity. 

In keeping with the results on young adults [8,9] and the “continuity hypothesis” [6,140], it has been recently demonstrated that the dream recall is also predicted by frontal theta oscillations in older adults (mean age = 69.2 years) [139].

Nightmares and bad dreams in older people are scarcely investigated. Several studies found a general decline in dream recall frequency during aging also for nightmares [216,217,218]. Specifically, Schredl and Gӧritz [217] analyzed the changes in dream recall rate and nightmare frequency in a large age range sample (16–89 years of age), over a three-year period. They found a reduction during aging, confirming findings from previous cross-sectional studies [219,220,221]. We could hypothesize that the decrease of strain and pressure in the elderly’s lifestyle could explain this phenomenon [217] since the nightmares are usually elicited by stressful factors [222]. 

On the other side, we must consider that nightmares are frequently associated with symptoms of anxiety [223]. In this respect, Nadorff and colleagues [224] confirmed a relation between generalized anxiety disorder (GAD) and the frequency of bad dreams in the elderly. Older adults with GAD showed significantly more bad dreams than healthy older subjects [223]. Although the prevalence of nightmares in the elderly is just equal to 4% [216], older adults that suffer from anxiety disorders frequently report nightmares [223], showing, once again, a continuity between sleep and waking experiences [6]. 

## 3. Limitations

It should be considered that most of the current findings on dreaming are collected from adult samples (age range of 18–47), while studies on children and older people are still in their early stages, especially concerning the electrophysiological approach. It should be considered that it has not been possible to review studies on children and adolescents separately, nor it has been possible to differentiate older adults in sub-groups, since most investigations have involved relatively wide age ranges, encompassing different age brackets.

Concerning infancy and childhood, the reasons are likely linked to the specific limitations associated with these age bracket, such as (a) the presence of infantile amnesia before 3-4 years of age [64,68]; and (b) the difficulty in collecting reliable dream reports from children because of their inability to discern dreaming from reality (the period in which this skill is acquired is yet to be defined) [66,68,69]. One of the main problems might be the compliance in collecting dream reports since the children may feel uncomfortable during an experimental session with unknown people. Moreover, when dreams are collected at home by questionnaire or diaries, children reports could be affected by the presence of parents. Both conditions can lead to a bias in collecting mental sleep activity.

Conversely, the lack of studies on dream experience in older adults could be ascribed to the relatively recent attention from the scientific community on this issue. Moreover, the elderly might be less willing to change their habits to sleep outside in a laboratory setting or to collect their dreams systematically.

## 4. Conclusions

In this review, we have summarized the available literature concerning changes in sleep and dream experiences across the lifespan and the relation between the oneiric activity and stressful or traumatic events, focusing on nightmares and disturbing dreams (PTSD-related). 

Although studies are still scarce and non-homogeneous among the different age brackets—both from neural and cognitive perspectives—some considerations should be advanced. 

Firstly, findings converge in showing that dream features mirror the cognitive functioning of subjects across the lifespan [74,78,80]. We hypothesize that dreaming is an expression of brain maturation, and the development of specific sleep patterns (e.g., maturational changes of sleep spindles and SWA) is strictly linked with cognitive functions and cortical plasticity [56,58,59,60,115,116,117,118,119,225].

It has been emphasized that visuo-spatial abilities and executive functions play a pivotal role in dream experience both in adults and in children (e.g., [74,78,80,136]), according to what has been observed by Solms on patients with brain lesions [12,13]. Moreover, memory could strongly impact on dreaming across the whole lifespan, since the dream cannot be recalled until the ability to organize the memories is developed [65]. Older adults remember dream contents related to key autobiographical memories more easily [210,211]. They also report some impoverished dream features [206] and a dramatic drop in dream recall rate when brain damage occurs [207]. 

From the electrophysiological perspective, theta oscillations both in young and older adults are related with the retrieval of dream contents [8,9,10,139], according to the view that the brain mechanisms promoting dream recall and memory processes are quite the same [140]. Considering the role of theta in the cortical maturation [38], the electrophysiological correlates of dreaming should also be tested during childhood.

Although no studies directly assessed the relationship between cognitive functioning and disturbing dreams during the lifespan, the literature points to a specific role of mental sleep activity in facing adverse or traumatic events. Bad dreams and nightmares could be attempts to cope with stressful events at any age. Periods characterized by life-changes appear to be more related to disturbing dreams in children [86]. Besides, studies on young adults found attenuation of the emotional charge related to wakefulness memories when the experiences are incorporated in dreams [170]. Older adults affected by anxious symptoms during wakefulness reported higher nightmare rates [224], while reduced disturbing dreams were observed during aging, in parallel to an attenuation of strain in the elderly’s life [222].

We can speculate that nightmares during the first childhood may be an unsuccessful attempt to cope with adverse events. Children might not have the abilities to reshape the unpleasant dream contents and regulate the emotional distress related to waking adverse events. It should be considered that such strategies could request complex cognitive functions (e.g., mental imagery) that may not be acquired during infancy. In keeping with the perspective that dreaming can represent an expression of human cognitive development, the prevalence of spontaneous lucid dreams is higher from adolescence until the early years of adulthood, as suggested by the “maturational hypothesis” by Voss and colleagues [226,227]. Interestingly, lucid dream, representing a greater awareness of own dream contents, helps narcoleptic adults to reduce their own nightmares [182]. This latter evidence is partly consistent with the view that the interest in the oneiric materials results in higher dream recall frequency [211,213]. In light of the above, it could be hypothesized that better awareness and interest in dreams may represent a useful strategy to attenuate negative items contained in the mental sleep activity. Some therapeutic protocols have been applied both in children and adults to ameliorate nightmares through a mental imagery technique [228,229,230]. The subjects were instructed to produce an alternative/positive scene to replace the negative emotional contents of the recurrent bad dreams.

We have also highlighted that “healthy” dreams, as well as disturbing dreams, are correlated with waking experiences (e.g., [10]). The “continuity hypothesis” has been supported by results provided from all age brackets (e.g., [10,80,139]). Although still scarce, investigations on nightmare sufferers provide evidence in favor of the hypothesis that brain activity during sleep is similar to that observed during the waking state for what concerns memory processes [89]. In particular, frontal theta oscillations are involved not only in the mechanisms aimed to remember dream contents in healthy subjects [8,139,140] but are also engaged in the modulation of abnormal dreams [89] and have been shown to “protect” individuals from developing PTSD [168].

While in healthy people dreaming may subserve memory consolidation processes [10], in PTSD or nightmares sufferers as well as in “stressed” people, it could promote the simulation of a new reality, where the daytime-traumatic contents would be modified or integrated with coping strategies and items that help problem-solving mechanisms [231]. Although fragmented results provide evidence in this direction, no studies on the electrophysiological correlates of this mechanism are available.

Future investigations are necessary to fill the gap in the literature on the neural correlates of dream recall and its alterations across the lifespan, also considering children and older samples. The detection of specific electrophysiological changes in relation to disturbing dreams could be considered a starting point for designing clinical protocols aimed to modify dream contents or the ability to recall/non-recall them. In this perspective, promising results from Voss et al. [154] showed that transcranial alternating current stimulation (tACS) may induce lucid dreams by a frontal gamma stimulation.

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
