# Peer review of "Mental Sleep Activity and Disturbing Dreams in the Lifespan"

_ijerph, 2019, doi:10.3390/ijerph16193658_

Round 1
Reviewer 1 Report
This in an interesting review of the literature supporting the view that dreaming can be considered an index of cognitive development across the lifespan. I think it needs some further work and English language editing but I look forward to reviewing the authors' revisions.
1) While many
of the mentioned studies cite participants' ages, it would be
better to have the age ranges discussed (e.g. children, adolescents and
elderly) early on in the review. How old is an elderly person? While it
is difficult to generalize on all types of individuals who belong to these age ranges, it would be useful to clearly
mark an age range within which these findings exist. There can be
significant variance across the ages of each of these stages of
development, especially in younger children and adolescents.
2) In
the section "Dream Recall In Older Adults". Despite being titled "older
adults", this section is quite broad in terms of the subjects
included in reviewed studies, and some of it refers to strictly the elderly
and psychopathology. One of the main studies discussed is from 10-79
years, not older adults. The arbitrary title of "older adults" doesn't
hold enough merit as representing thematically the studies in this
section. I think this can be reworked so to find a better theme for this
section, or to fit parts of this section under other sections.
3) A
number of times ideas were summarised from what was previously written,
but new citations referring to this information were included as if new. For example, if the findings from [171] are being put into other words, why is
this rewording using a different citation of [172]. Perhaps use the
citation elsewhere to avoid merging similar studies but framing
them as having found the same thing.
Minor observations
Line 31: This could be expanded upon further, to explain what the interval of dream production means.
Line 57: "the" should be omitted here
Line 72: missing an ) after [11]
Line 142: The authors have just explained that dream recall starts from 3-4 years (Line 137), but these studies mention recall from 2-5. Please provide explanation for difference / methods, ensuring validity.
Line 248: Please omit "the" before adolescence. Furthermore, it is a bold claim to make when considering how plastic the brain is. Perhaps avoid using the term "finish" strictly in adolescence.
Lines 280 - 284: Please break down into separate sentences as this is hard to follow
Line 322 onwards: There seems to be a slight change in writing style, please uniform with the preceding section (for example, definitions in this manuscript were previously given outside of the brackets and now included, see lines 334-335).
Line 333: in the ongoing brain operations of what? Wakefulness?
Line 344: Less faster should be less fast, or slower
Line 357: please break the sentence down after [95]
Lines 358 - 362: Hard to follow, please break down
Line 389: I don't think DE has been defined previously.
Line 392: Please clarify what is meant by (stressful) events.
Line 399: Please change "respect" to "as compared to a" 5% prevalence
Lines 410-413: perhaps break up into two sentences for legibility.
Lines 417-418: "About that" sounds too colloquial, please modify
Line 422: It would be good to have the "Elderly" concept operationalised and have ages from which these patterns may arise.
Lines 436: Please change "maximal" with "most prominent"
Line 444: Should read "A significant reduction of ..... has been reported in middle-age and older adults"
Lines 453-458: break up into two sentences for legibility.
Line 504: DR has not previously been defined
Line 508: Please move Grenier et al. 222 before "recorded"
Lines 519-520: Please move to the previous paragraph.
Author Response
This in an interesting review of the literature supporting the view that dreaming can be considered an index of cognitive development across the lifespan. I think it needs some further work and English language editing but I look forward to reviewing the authors' revisions.
Thanks to the reviewer for her/his appreciation. According to the request, we checked for typos and grammar errors, and the manuscript has been reviewed by a native English speaker. However, if you believe that the revised manuscript needs of a further language revision, we will use a professional editing service.
1) While many of the mentioned studies cite participants' ages, it would be better to have the age ranges discussed (e.g. children, adolescents and elderly) early on in the review. How old is an elderly person? While it is difficult to generalize on all types of individuals who belong to these age ranges, it would be useful to clearly mark an age range within which these findings exist. There can be significant variance across the ages of each of these stages of development, especially in younger children and adolescents.
The reviewer is right. Since the scarcity of dream studies in children and adolescents, it has not been possible providing different sections for these two age ranges. Moreover, some studies reported results from sample including wide age range (e.g, from pre-scholar age to 14/16 years). In order to clarify this issue, we have inserted in the introduction the age range considered in each section. Also, we have mentioned in a new section “Limitations” the issue concerning a relative overlap among the age ranges of some different sections.
2) In the section "Dream Recall In Older Adults". Despite being titled "older adults", this section is quite broad in terms of the subjects included in reviewed studies, and some of it refers to strictly the elderly and psychopathology. One of the main studies discussed is from 10-79 years, not older adults. The arbitrary title of "older adults" doesn't hold enough merit as representing thematically the studies in this section. I think this can be reworked so to find a better theme for this section, or to fit parts of this section under other sections.
We understand the issue raised by the reviewer. However, we have to underline that the results mentioned in this section concern the changes of dreaming characteristics in older adults compared to younger sample. Hence, some cross-sectional and/or longitudinal studies include –necessarily- younger groups. We have better clarified this point in the section.
3) A number of times ideas were summarised from what was previously written, but new citations referring to this information were included as if new. For example, if the findings from [171] are being put into other words, why is this rewording using a different citation of [172]. Perhaps use the citation elsewhere to avoid merging similar studies but framing them as having found the same thing.
We have tried to change introduction and discussion according to the right request of the reviewer. We sincerely hope that this problem is now overpassed.
Minor observations
Thanks so much for all these helpful suggestions.
Line 31: This could be expanded upon further, to explain what the interval of dream production means.
Done.
Line 57: "the" should be omitted here
Done.
Line 72: missing an ) after [11]
Done.
Line 142: The authors have just explained that dream recall starts from 3-4 years (Line 137), but these studies mention recall from 2-5. Please provide explanation for difference / methods, ensuring validity.
Corrected.
Line 248: Please omit "the" before adolescence. Furthermore, it is a bold claim to make when considering how plastic the brain is. Perhaps avoid using the term "finish" strictly in adolescence.
Done.
Lines 280 - 284: Please break down into separate sentences as this is hard to follow
Done.
Line 322 onwards: There seems to be a slight change in writing style, please uniform with the preceding section (for example, definitions in this manuscript were previously given outside of the brackets and now included, see lines 334-335).
Done.
Line 333: in the ongoing brain operations of what? Wakefulness?
Specified.
Line 344: Less faster should be less fast, or slower
Corrected.
Line 357: please break the sentence down after [95]
Done.
Lines 358 - 362: Hard to follow, please break down
Done.
Line 389: I don't think DE has been defined previously.
Corrected.
Line 392: Please clarify what is meant by (stressful) events.
Specified.
Line 399: Please change "respect" to "as compared to a" 5% prevalence
Done.
Lines 410-413: perhaps break up into two sentences for legibility.
Done.
Lines 417-418: "About that" sounds too colloquial, please modify
Changed.
Line 422: It would be good to have the "Elderly" concept operationalised and have ages from which these patterns may arise.
Done.
Lines 436: Please change "maximal" with "most prominent"
Done.
Line 444: Should read "A significant reduction of ..... has been reported in middle-age and older adults"
Done.
Lines 453-458: break up into two sentences for legibility.
Done.
Line 504: DR has not previously been defined
Corrected.
Line 508: Please move Grenier et al. 222 before "recorded"
Done.
Lines 519-520: Please move to the previous paragraph.
Done.
Reviewer 2 Report
In this review article titled “Dreaming in the lifespan: the role in cognitive processes and sleep disorders” the authors have tried to establish a link between how REM sleep changes with age and how it is related to the neurological disorders. In totality, the information gathered in this article can be great for the reader. However, the article needs a through re-writing keeping the following points in mind.
Major:
1. Professional English checking is needed.
2. The information listed is fragmented and sometimes make no meaning. At many places, factual information from original articles has also been listed in the article. In the reviewer’s opinion, the authors need to write in a conclusive way, too many actual results from the original research articles disrupt the flow of the information and make it difficult to understand. In short, it is not necessary to include information from each and every paper available in the literature.
3. There are too many small-small paragraphs, the main sections are too long. Bigger sections can be divided into small subsections with a clear heading so that the reader has an understanding of the content.
4. Conclusions need to be re-written. It’s too long and out of focus, no actual conclusions are found in it.
5. The whole section 2.3 Elderly people needs to be reorganized. It seems more like a listing of all published work in this area, instead of real distilling of published work into conclusions.
6. It feels like the actual topic of the review (role of dreaming in cognitive processes and sleep disorders) has been overlooked. The title neither the abstract fits the contents of the manuscript. Authors had made a commendable work putting together a vast amount of information but again it’s necessary to reorganize the sections and integrate all the data in key ideas.
Minor
1. Page 5 lines 198-202. Please confirm the adequacy of these references. In the study from Hawkins et al, it was not reported nightmares per week, instead, they report at least once per month or per 6 months
2. Page 6 line 273. Authors overlook the fact that the correlation between sleep spindles activity and IQ was found in females. Please discuss the sexual dimorphism further along the review.
3. Page 4 line 192. Please specify: Dream experience and sleep disorders in children
4. Page 8 line 354. Please specify: Dream experience in sleep disorders in young adults
5. Page 8 line 371-372. The following sentence is vague: “In other words, … a greater activation…nightmares”. Please specify which activation the authors are referring to.
6. Page 11 line 539. Please specify: Dream experience in sleep disorders in older adults
7. Some abbreviations along the review need to be defined (i.e. DE, DR,..).
8. Reference: Need to be unified, many of them include DOI while others don’t.
Author Response
Thanks to the reviewer for her/his suggestions.
Professional English checking is needed.According to the suggestions of the reviewers, we checked for typos and grammar errors, and the manuscript has been reviewed by a native English speaker. However, if you believe that the revised manuscript needs of a further language revision, we will also use a professional editing service.
The information listed is fragmented and sometimes make no meaning. At many places, factual information from original articles has also been listed in the article. In the reviewer’s opinion, the authors need to write in a conclusive way, too many actual results from the original research articles disrupt the flow of the information and make it difficult to understand. In short, it is not necessary to include information from each and every paper available in the literature.According to the suggestions, we have removed “factual information” along the manuscript.
There are too many small-small paragraphs, the main sections are too long. Bigger sections can be divided into small subsections with a clear heading so that the reader has an understanding of the content.According to the suggestions, we have divided the bigger “conclusions” section in two (3. Limitations and 4. Conclusions). Also, the section 2.3 has been reorganized and the smallest paragraph (2.3.3) was incorporated in a paragraph 2.3.2.
Conclusions need to be re-written. It’s too long and out of focus, no actual conclusions are found in it.The section has been reorganized.
The whole section 2.3 Elderly people needs to be reorganized. It seems more like a listing of all published work in this area, instead of real distilling of published work into conclusions.The section has been reorganized and the paragraphs 2.3.2 and 2.3.3 have been inserted within a single paragraph.
It feels like the actual topic of the review (role of dreaming in cognitive processes and sleep disorders) has been overlooked. The title neither the abstract fits the contents of the manuscript. Authors had made a commendable work putting together a vast amount of information but again it’s necessary to reorganize the sections and integrate all the data in key ideas.According to the reviewer suggestions, the title and the abstract has been change to better fit the contents of the manuscript. Also, the abstract has been revised. Finally, the conclusions have been rewritten, and a brief paragraph on limitations has been inserted in the revised manuscript.
Minor
Page 5 lines 198-202. Please confirm the adequacy of these references. In the study from Hawkins et al, it was not reported nightmares per week, instead, they report at least once per month or per 6 months.Thanks. It has been corrected.
Page 6 line 273. Authors overlook the fact that the correlation between sleep spindles activity and IQ was found in females. Please discuss the sexual dimorphism further along the review.The reviewer is right and we have clarified. On the other hand, discussing the issue of sexual dimorphism in dream recall is out of the topic of this review, since it is a general (and relevant) topic which deserves an indipendent and specific review.
Page 4 line 192. Please specify: Dream experience and sleep disorders in childrenChanged.
Page 8 line 354. Please specify: Dream experience in sleep disorders in young adultsChanged.
Page 8 line 371-372. The following sentence is vague: “In other words, … a greater activation…nightmares”. Please specify which activation the authors are referring to.Done.
Page 11 line 539. Please specify: Dream experience in sleep disorders in older adultsChanged.
Some abbreviations along the review need to be defined (i.e. DE, DR,..).Corrected.
Reference: Need to be unified, many of them include DOI while others don’t.Done.
Reviewer 3 Report
needs to be rewritten in clear, grammatically correct English - very hard to understand what you are proposing in the paper.
Author Response
According to the request, we checked for typos and grammar errors, and the manuscript has been reviewed by a native English speaker. However, if the Editor and the reviewers thinks that the revised manuscript needs of a further language revision, we will use a professional editing service (i.e., much delaying the final version of this review).
Round 2
Reviewer 3 Report
some errors are still present and there is some imprecision around terms that I feel is crucial to the manuscript's argument - would suggest a rewrite with an editor